# Crude Lipopeptides Produced by *Bacillus amyloliquefaciens* Could Control the Growth of *Alternaria alternata* and Production of *Alternaria* Toxins in Processing Tomato

**DOI:** 10.3390/toxins16020065

**Published:** 2024-01-25

**Authors:** Yuanyuan Zhang, Yingying Fan, Yingying Dai, Qinlan Jia, Ying Guo, Peicheng Wang, Tingting Shen, Yan Wang, Fengjuan Liu, Wanhui Guo, Aibo Wu, Ziwei Jiao, Cheng Wang

**Affiliations:** 1College of Biological Sciences and Technology, Yili Normal University, Yining 835000, China; zhangyuanyuan9990@163.com (Y.Z.); gying927@163.com (Y.G.); 2Institute of Quality Standards & Testing Technology for Agro-Products, Xinjiang Academy of Agricultural Sciences/Key Laboratory of Functional Nutrition and Health of Characteristic Agricultural Products in Desert Oasis Ecological Region (Co-Construction by Ministry and Province), Ministry of Agriculture and Rural Affairs/Laboratory of Quality and Safety Risk Assessment for Agro-Products (Urumqi), Ministry of Agriculture and Rural Affairs/Key Laboratory of Agro-Products Quality and Safety of Xinjiang, Urumqi 830091, China; fyyxaas@xaas.ac.cn (Y.F.); dyyhj999@sina.com (Y.D.); jiaql9813@163.com (Q.J.); wangyan@xaas.ac.cn (Y.W.); liufengjuan@xaas.ac.cn (F.L.); guowanhui0504@126.com (W.G.); 3College of Life Science and Technology, Xinjiang University, Urumqi 830049, China; 4College of Food Science and Pharmacy, Xinjiang Agricultural University, Urumqi 830052, China; m13565903043@163.com (P.W.); m18799666994@163.com (T.S.); 5CAS Key Laboratory of Nutrition, Metabolism and Food Safety, Shanghai Institute of Nutrition and Health, Shanghai Institutes for Biological Sciences, University of Chinese Academy of Sciences, Chinese Academy of Sciences, Shanghai 200031, China; abwu@sibs.ac.cn

**Keywords:** *Bacillus amyloliquefaciens*, lipopeptide, *Alternaria alternata*, processing tomato, *Alternaria* toxins

## Abstract

*Alternaria* spp. and its toxins are the main contaminants in processing tomato. Based on our earlier research, the current study looked into the anti-fungal capacity of crude lipopeptides from *B. amyloliquefaciens* XJ-BV2007 against *A. alternata*. We found that the crude lipopeptides significantly inhibited *A. alternata* growth and reduced tomato black spot disease incidence. SEM analysis found that the crude lipopeptides could change the morphology of mycelium and spores of *A. alternata*. Four main *Alternaria* toxins were detected using UPLC-MS/MS, and the findings demonstrated that the crude lipopeptides could lessen the accumulation of *Alternaria* toxins in vivo and in vitro. Meanwhile, under the stress of crude lipopeptides, the expression of critical biosynthetic genes responsible for TeA, AOH, and AME was substantially down-regulated. The inhibitory mechanism of the crude lipopeptides was demonstrated to be the disruption of the mycelial structure of *A. alternata*, as well as the integrity and permeability of the membrane of *A. alternata* sporocytes. Taken together, crude lipopeptides extracted from *B. amyloliquefaciens* XJ-BV2007 are an effective biological agent for controlling tomato black spot disease and *Alternaria* toxins contamination.

## 1. Introduction

Tomato (*Solanum lycopersicum*) is the second most produced vegetable in the world and is rich in nutritious bioactive substances [1]. Xinjiang is located at 41° N 85° E in Northwest China. It has grey desert soil and aeolian sandy soil as the primary agricultural soil types, and a semiarid or desert climate with great seasonal differences in temperature [2,3]. One of the most suitable locations worldwide for planting tomatoes is Xinjiang because of the fertile soil, lots of sunshine, and significant day-to-night temperature variation. The top 10 tomato producers are as follows: China, India, USA, Turkey, Egypt, Iran, Italy, Spain, Mexico, and Brazil [4]. The output from processing tomatoes accounts for 80% of the total output of the country, and the output of tomato paste accounts for 90% of the total processing in the country [5]. Tomatoes are frequently infected in the field, during harvest, and in storage by *Alternaria* spp. [6]. Spores of *A. alternata* have been detected in both contaminated soil and tomatoes, which cause black and concave spots on a tomato fruit, not only reducing the yield but also affecting the quality and nutritional value of the tomato.

*Alternaria* spp. produce about 70 metabolites, namely, *Alternaria* toxins, with apparent harmful effects on people and animals, among which Tenuazonic acid (TeA), Alternariol (AOH), Alternariol methyl ether (AME), Tentoxin (TEN), and Altenuene (ALT) are familiar to us. The production of *Alternaria* toxins is associated with some key genes. *A. alternata* with an inactivated *aahog* gene was shown to be incapable of producing AOH, suggesting that the *aahog* gene is necessary for producing AOH [7]. Reduced expression of *pksI* and delayed generation of AOH were caused by the deletion of the transcription factor gene *aohR* [8]. Yun et al. discovered that *TAS2* and *PoLAE1*, two distinct regulators, control the production of TeA in *Pyricularia Oryzae* [9]. *AaTAS1* and *AaMFS1* were involved in the biosynthesis and metabolic regulation of TeA, according to Sun et al. [10].

In recent decades, scientists have studied the toxicity of *Alternaria* toxins without interruption. According to reports, food contaminated by *A. alternaria* may have a direct correlation with the high rate of esophageal cancer [11]. Liu et al. found that the mutagenicity and carcinogenicity of AOH and AME were linked with human esophageal cancer lesions [12]. The US Food and Drug Administration Register of Toxic Chemicals lists TeA as the most toxic *Alternaria* toxins [13]. TeA is cytotoxic and causes dizziness, tachycardia, and gastrointestinal bleeding in mammals, leading to death. Moreover, it is thought to be responsible for Onyalai, a human blood disorder [14,15]. In testing the toxicity of *Alternaria* toxins using human gastric epithelial cell lines, Lin et al. demonstrated that AOH + AME + TeA had synergistic toxic effects that activated caspase-3 cleavage, induced apoptosis, and activated the DNA damage pathways ATM-Chk2-P53 and ATR-Chk1-P53 [16].

The contamination of *A. alternata* has been detected in fruits [17,18], vegetables [19], cereals, and their products [20,21]. Tomato and its products are one of the critical targets for contamination by *Alternaria* toxins. For example, *Alternaria* toxins were detected in Brazilian tomatoes and their derivatives, where 68.4% of the original samples were contaminated with *A. alternata* [22]. Sanzani et al. analyzed dried and fresh tomato samples from Southern Italian markets and processing factories for toxins, and they found that all of the samples were contaminated with TeA, with AME coming up second, then TEN, and AOH [23]. According to a market survey conducted in Switzerland, TeA (95%) was most commonly found in tomato-based goods at levels as high as 790 µg/kg, whereas AOH and AME quantities varied between 1 and 33 µg/kg [24]. The primary toxins found in all tomato juice and paste samples randomly collected from various locations in the Chinese market were found to be TeA [25]. Long-term consumption of toxin-contaminated products poses a hazard to the well-being of humans. Controlling *A. alternata* growth and the buildup of *Alternaria* toxins in tomatoes is, therefore, essential.

At present, there are mainly two methods of the prevention and control of mycotoxins. One is to slow down the growth of toxin-producing fungi, thus reducing mycotoxin synthesis and the other is the degradation of mycotoxins. For the above first method, *Bacillus* spp., as the antagonistic bacteria, are commonly used, and lipopeptides produced by them have been proven to be their main fungistatic substances. As a preservative, for instance, a few of the lipopeptides made by *B. subtilis* can stop the growth of phytopathogenic microbes in food [26]. Based on their structure, lipopeptides could be categorized into the iturin family, the surfactin family, and the fengycin family. Cozzolino et al. isolated a bacterial strain *Bacillus* SL-6 from freshwater in a reservoir, which was demonstrated to secrete fengycin that possesses anti-fungal activity, and it completely suppressed the rot in apples caused by *A. alternata* [27]. The anti-fungal capacity of *B. amyloliquefaciens* PG12 isolated from apple fruits could control apple ring rot, and it was associated with the effect of lipopeptides, in which iturin A plays a vital role in influencing this process [28]. Lipopeptides secreted by *B. amyloliquefaciens* BPD1 were antagonistic to *Pyricularia oryzae* Cavara (PO), and these lipopeptides caused damage and tumor formation in the hyphae and conidia of PO [29]. However, these studies have been limited to the inhibition of pathogenic fungi and have not been reported for toxins.

In our previous study, *B. amyloliquefaciens* XJ-BV2007 could inhibit tomato black spot disease pathogen *A. alternata* isolate H10, and it was preliminary explored that crude lipopeptides were the primary antibacterial active substance. Still, the inhibitory mechanism of lipopeptides on *A. alternata* has not been studied [30]. In actual agricultural activities, bacterial agents were not a good choice for their unstable proliferation and vulnerability to fungicides [31]. Therefore, it is expected to use crude lipopeptides as preventive and control agents, which are easy to extract and resistant to heat, acid, and alkali. The goals of the present investigation were to assess the impact of crude lipopeptides from *B. amyloliquefaciens* XJ-BV2007 on the control for *A. alternata* isolate H10 and *Alternaria* toxins with in vivo and in vitro assays, and to explore the possible mechanisms of crude lipopeptides in controlling the postharvest black spot disease. The results of this study will offer fresh perspectives on applying crude lipopeptides as a bio-control agent for controlling tomato black spot disease and *Alternaria* toxin contamination. This will be extremely valuable and significant for the further growth of the tomato processing industry.

## 2. Results and Discussion

### 2.1. Effect of Crude Lipopeptides Treatment on Mycelial Growth of A. alternata

Through confronting incubation on a PDA medium, the anti-fungal activity of crude lipopeptides against *A. alternata* was assessed. The radial growth for *A. alternata* upon PDA mediums had been impeded by crude lipopeptides with fungal growth inhibition of up to 56% (Figure 1a). With increased incubation time, crude lipopeptides’ ability to inhibit *A. alternata* was progressively enhanced. Meanwhile, the mycelium in the mediums with the addition of crude lipopeptides had a lighter colony color (Figure 1b) than the control group (CK). Biomass is one of the most significant indicators of mycelial growth and reflects the accumulation of substances during growth and metabolism [32]. As shown in Figure 1c, the biomass of *A. alternata* was reduced by 18.5%, 28.2%, 42.6%, 49.3%, 53.6%, 57.5%, and 60.1%, respectively, compared with the CK.

The production of antimicrobial compounds has been recognized as one of the modes of antagonistic action. Numerous *Bacillus* spp. have been shown to be helpful for the biocontrol of a variety of fungal pathogens. According to Zhang et al., *B. subtilis* JK-14 could lower the incidence of fungal development on peaches and the mycelial growth of *A. tenuis* in a PDA medium [33]. The mycelial growth for *A. solani* was crushed by *B. amyloliquefaciens* TF28 with a percentage of inhibition over 90% [34]. Thus, in this study, *A. alternata*’s growth was inhibited by the crude lipopeptides of *B. amyloliquefaciens* XJ-BV2007.

### 2.2. Stability Test of Anti-Fungal Activity of Crude Lipopeptides

*Bacillus* spp. are unique in their capacity for quick replication, tolerance to harsh environmental conditions, and an extensive spectrum of biocontrol abilities [35]. In Figure 2, the stability of crude lipopeptides was related to the growth diameter of *A. alternata*. Thermal stability results showed that the diameter of *A. alternata* did not change after the incubation of crude lipopeptides treated at 25 °C and 60 °C. However, *A. alternata*’s diameter increased at 120 °C, indicating that the activity of the crude lipopeptides was unaffected by different heat treatments below this temperature (Figure 2a). The unstable climate change may affect the levels of *A. alternata* and mycotoxins in tomatoes [36]; thus, the stable lipopeptides might a good choice.

Tomato plants grow well in well-drained sites that receive full sun most of the day. The soil pH should be slightly acidic (6.2 to 6.8) [37]. The findings of the pH stability experiment showed that the crude lipopeptides were stable over a range of pH conditions. At a pH of 1.0 to 9.0, the activity of crude lipopeptides was unaffected, while at a pH of 11.0 to 13.0, the activity of crude lipopeptides decreased, and the *A. alternata* radial diameter increased (Figure 2b). A previous research found that the lipopeptides produced by *B. subtilis* SYL-6 had a stable property and a satisfactory antimicrobial effect [38]. As a result, the crude lipopeptides were sufficiently stable for use in agricultural applications.

### 2.3. Inhibition Mechanism of Crude Lipopeptides on A. alternata

#### 2.3.1. Observation of Mycelium and Spores Microstructure

The structural change of *A. alternata* was observed via SEM. The mycelial in the CK group appeared fine folds on the surface, slightly twisted, and showed a round and plump cylinder (Figure 3a). After treatment with crude lipopeptides, the mycelium of *A. alternata* collapsed and appeared ruptured, losing its cylindrical structure (Figure 3b).

The morphology of *A. alternata* spores was significantly altered by crude lipopeptides. The spores in the CK group had fine folds on the surface and a little sunken phenomenon (Figure 3c). While treated with crude lipopeptides, spores were twisted, resulting in a noticeable sunken phenomenon as well as a significant crack on the surface of the spore (Figure 3d). It is owing to an effect of crude lipopeptides in the outer cell wall and cell membrane of *A. alternata*, destroying its original integrity and normal metabolic function, leading to cytoplasmic leakage and hindering the regular growth and development of mycelium and spores [39].

#### 2.3.2. Effect of Crude Lipopeptides Treatment on the Membrane Integrity of *A. alternata* Spore Cells

Propidium iodide (PI) is a nuclear staining reagent that penetrates broken cell membranes, binds to intracellular nucleic acids, stains the cell nucleus, and fluoresces in cytoplasmic material [40]. The strength of the fluorescence indicates the cells integrity, and the higher fluorescence intensity indicates that the cell membrane is more severely damaged. As shown in Figure 4b, there was no red fluorescence observed in the CK group, indicating that the spores and cell membranes were intact and undamaged. Nevertheless, after treatment with crude lipopeptides, the cell membranes in most *A. alternata* spores were disrupted and emitted strong red fluorescence, indicating that crude lipopeptides severely harmed the structure of the cell membranes for *A. alternata* spores, with a damage inhibition rate of 85% (Figure 4a). This observation bears similarities to those made by Kaur et al. [41], who observed that PI could be stained in the area of the swollen and vacuolated anomalies in *A. alternata* using iturin A extracted from *B. vallismortis*.

#### 2.3.3. Effect of Crude Lipopeptides Treatment on Conductivity of *A. alternata* Mycelium

Biofilms are crucial elements in maintaining cell structure and physiology [42]. Mycelium extracellular conductivity is an essential indicator of cell membrane permeability. As shown in Figure 5, the conductivity of mycelium was increased over time in all treatments compared to the control group. After the crude lipopeptides treatment, the conductivity of *A. alternata* mycelium tended to increase at different treatment times, and it was substantially higher than that in the CK group due to electrolyte exosmosis within mycelial cells. The conductivity of *A. alternata* mycelium at 180 min after crude lipopeptides treatment was 1.50 times higher than that in the CK group. The result was in line with previous findings, in which the anti-fungal compounds iturin A2 as well as iturin A6 of *B. safensis* B21 impeded the growth of pathogenic fungi by disrupting the permeation of fungal cell membranes [43]. In this study, it was found that crude lipopeptides treatment immediately increased the permeability of fungal cell membranes, resulting in the efflux for intracellular ions as well as the elevation of extracellular conductivity.

### 2.4. Effect of Crude Lipopeptides on the Content of Alternaria Toxins

To further investigate the inhibitory ability of crude lipopeptides on *Alternaria* toxins, UPLC-MS/MS was selected to detect their content. As shown in Figure 6, crude lipopeptides had a substantial inhibitory impact on the growth of *A. alternata* in the PDB medium when compared with the CK. It is noteworthy that an accumulation of TeA, AOH, AME, and TEN of *Alternaria* toxins was reduced by 99%, 46%, 95%, and 63%, respectively, by the addition of crude lipopeptides compared with the CK (Figure 7).

*Bacillus* spp. could produce secondary metabolites with high antagonistic activity and is considered an ideal candidate for biocontrol [44]. Arrebola et al. showed that iturin A and bacillomycin D present in the supernatant were the causes of *B. amyloliquefaciens* PPCB004’s antagonism toward *P. expansum* [45]. According to Zouari et al., lipopeptides of *B. amyloliquefaciens* CEIZ-11 demonstrated an extensive range of anti-fungal action toward different plant-borne pathogenic fungi [46]. Hu et al. found that selenium nanoparticles significantly improved control mechanisms against *Alternaria* toxins and exhibited better anti-fungal activities [47]. In summary, it was proposed that crude lipopeptides could hinder the growth of *A. alternata* as well as control its production of *Alternaria* toxins.

### 2.5. Effect of Crude Lipopeptides on A. alternata and Its Toxins on Tomato

In Figure 8a, after a week of storage, the mean lesion diameter of the tomatoes inoculated solely with *A. alternata* was 30.8 mm. Conversely, the mean lesion diameter of the tomatoes treated with crude lipopeptides and *A. alternata* was 3.09 mm, representing an 89.97% reduction in the lesion diameter (*p* < 0.01). In Figure 9, after vaccination with *A. alternata*, the contents of TeA, AOH, AME, and TEN in crude lipopeptides-treated processing tomato were less than those of the control group by 100%, 98%, 99%, and 53%, respectively (*p* < 0.01).

Since *B. subtilis* is widely distributed and can produce an extensive variety of metabolites, it is a strong contender to be used as a biocontrol agent to manage plant diseases [48]. Numerous *B. amyloliquefaciens* species have been shown by certain writers to be effective in the biocontrol of a range of fungal phytopathogens. Ya’nez-Mendiza’bal et al. reported that the lipopeptides of *B. subtilis* CtpxS2-1 treatment completely controlled anthracnose incidence [49]. In the study by Xiong et al., the antagonistic strain *Aspergillus oryzae* M30011 showed a break down of ochratoxin A (OTA) to harmless OTα with a 94.0% degradation rate in 72 h [50]. The ε-poly-L-lysine treatment inhibited TEN content in *A. alternata* but promoted the content of AOH, AME, and ALT [51]. There has not been any research reported on how antagonistic microorganisms could inhibit *Alternaria* toxins in tomatoes thus far. In this study, the crude lipopeptides of *B. amyloliquefaciens* XJ-BV2007 effectively controlled postharvest black spot disease of tomato and inhibited a buildup of *Alternaria* toxins.

### 2.6. Effect of Crude Lipopeptides on the Expression of Key Genes Related to Alternaria Toxins

To understand the functions of crude lipopeptides in *Alternaria* mycotoxins formation, gene expression levels related to *Alternaria* toxins biosynthesis were investigated. In Figure 10, while *A. alternata* was treated with crude lipopeptides, the relative expression of *omtI* and *aohR*, which are involved in AOH as well as AME biosynthesis, declined 26.90- and 4.35-fold, respectively. The production of AOH and AME may be directly decreased by down-regulating *omtI* and *aohR* [39]. Furthermore, after treatment with crude lipopeptides, it was observed that there was a significant down-regulation by the relative expression genomes of *AaMFS1* and *AaTAS1*, which are responsible for TeA biosynthesis, by 2.39-fold and 3.46-fold magnitudes, respectively, in comparison to the CK. A polyketide gene cluster in *A. alternata* is accountable for the biosynthesis of AOH and its derivatives, and *aohR* is a positive transcriptional factor that promotes the mycotoxin development [10]. Magnolol hindered the synthesis of AOH and AME by down-regulating a conveying for genes that are grouped around polyketides, such as *pksI* and *omtI* [39]. In summary, crude lipopeptides could control *Alternaria* toxins by inhibiting the expression of critical genes associated with toxin biosynthesis.

### 2.7. Discussion of the Possible Inventory Path to Alternaria Toxins

Collectively, the above results showed that crude lipopeptides have a significant inhibitory effect on *A. alternata*. Possible inhibitory pathways to *Alternaria* toxins of the crude lipopeptides were proposed as follows (Figure 11). The crude lipopeptides, extracted from *B. amyloliquefaciens* XJ-BV2007, had strong anti-fungal activity and significantly inhibited the growth of *A. alternata* isolate H10 on PDA, PDB, and tomatoes. It was shown the crude lipopeptides were capable of substantially inhibiting the morphology of *A. alternata* mycelial and spores via SEM analysis, in addition to damaging the integrity and permeability of the *A. alternata* isolate H10 cell membrane. Moreover, it also suppressed the expression of critical biosynthetic genes associated with *Alternaria* toxins, thereby inhibiting toxins production. Therefore, by inhibiting *A. alternata* growth and lowering its toxins level, *B. amyloliquefaciens* XJ-BV2007 and its crude lipopeptides could increase the safety of processing tomatoes that are prone to *A. alternata* contamination.

## 3. Conclusions

In this study, a significant inhibitory effect of crude lipopeptides on the growth of *A. alternata* was demonstrated through an anti-fungal experiment on a PDA medium. The SEM results showed that crude lipopeptides could inhibit the shape of *A. alternata* mycelial and spores, and they damaged the integrity and permeability of spore cell membranes. The ability of crude lipopeptides to inhibit the growth of *A. alternata* and the diameter of tomato spots was verified through the inhibition experiment on a PDB medium and the processing of tomatoes, and the accumulation of *Alternaria* toxins was reduced. Moreover, we discovered that crude lipopeptides decreased the generation of *Alternaria* toxins by inhibiting the expression of toxin biosynthetic genes. Thus, crude lipopeptides could be considered as a potential biocontrol agent to combat *A. alternata* which causes black spot disease and pollution of *Alternaria* toxins in processing tomatoes.

## 4. Materials and Methods

### 4.1. Microorganisms and Culture Conditions

*B. amyloliquefaciens* XJ-BV2007 was isolated from the fields of processing tomatoes in the Yanqi Basin of Xinjiang in our laboratory, and it was preserved in China General Microbiological Culture Collection Center as CGMCC No. 23669. It was then moved to a synthetic fermentation medium containing 11.25 g of peptone, 3.75 g of yeast extract powder, 5 g of soluble starch, and 1 g of NaCl per litre. It was shaken continuously for 36 h at 150 rpm/min within a rotary shaker over 28 °C.

The processing tomato black spot pathogen, *A. alternata* isolate H10, was isolated from diseased tomatoes and stored in −20 °C sterilized glycerol (35% (*v*/*v*)). It was activated in fresh potato dextrose agar (PDA) medium and incubated at a temperature of 28 °C for 7 days.

### 4.2. Chemicals and Reagents

Nutrient agar (NA) and nutrient broth (NB) were purchased from Qingdao Hope Bio-Technology Co., Ltd. (Qingdao, China). Potato dextrose agar (PDA) and potato dextrose broth (PDB) were provided by Beijing Land Bridge Technology Co., Ltd. (Beijing, China). LC-MS grade acetonitrile and acetic acid were supplied by Thermo Fisher Scientific (Waltham, MA, USA). Standard solutions of TeA, AOH, AME, and TEN were purchased from Romar Labs Division Holding GmbH (Getzersdorf, Austria). Propidium iodide (PI) solution was provided by Sigma-Aldrich, Co., Ltd. (Burlington, MA, USA). The Fungal RNA Kit was purchased from Omega Bio-tek, Inc. (Norcross, GA, USA). The FastKing Rt Kit (With gDNase) and SuperReal PreMix Plus (SYBR Green) were purchased from Tiangen Biotech Co., Ltd. (Beijing, China).

### 4.3. Instruments

XSE204 balance was provided by Mettler-Toledo (Greinfesee, Switzerland). A11 basic ultra-turrax and MS3 vortex mixer were purchased from IKA (Staufen, Germany). Automatic horizontal shaker was provided by Hannuo Instruments Co., Ltd. (Shanghai, China). The precision vertical thermostatic light oscillation incubator was provided by Laibo Technology Co., Ltd. (Tianjin, China). JSM-6390 LV scanning electron microscopy was purchased from Japan Electronics Co., Ltd. (Tokyo, Japan). U-LH100-3 fluorescence microscopy was acquired from Yonke Instruments Co., Ltd. (Shanghai, China). N60 nanophotometer was provided by Implen Gmbh (Munich, Germany). Quantitative real-time PCR was purchased from Roche Diagnostics GmbH (Mannheim, Germany).

### 4.4. Experimental Methods

#### 4.4.1. Extraction of Crude Lipopeptides

The extraction method of lipopeptide substances was based on that of Kim et al. [52], with a few modifications. The fermentation broth of *B. amyloliquefaciens* XJ-BV2007 was diluted to the concentration of 1 × 10^5^ spore mL^−1^ based on the A_600_ value and centrifuged at 20 °C, 10,000× *g* for 15 min. The resulting liquid was gathered as a cell-free filtrate [53]. Then, 300 mL cell-free filtrate was held at 4 °C overnight after being brought to a pH of 2.0 with HCl. The residue was obtained by centrifuging it at 10,000× *g* for 10 min at 20 °C and then extracted using methanol with three times the precipitate weight at least twice. After being dried at 40 °C in a rotary evaporator, it was filtered through hydrophilic PTFE bilayer membranes with a 0.22 µm. Finally, a yield of 2 mL crude lipopeptides was obtained, containing 4 µg mL^−1^ of the fengycin, which had been demonstrated to be the main lipopeptides in our previous work [30].

#### 4.4.2. Effect of Crude Lipopeptides on the Growth of *A. alternata*

A total of 5 mL of *B. amyloliquefaciens* XJ-BV2007 crude lipopeptides was mixed with the PDA solution at a ratio of 1:3. After solidification, sterile cellophane was carefully spread on the surface of the medium. *A. alternata* was inoculated for 7 days at 28 °C in the middle of the medium above. The colony diameter of *A. alternata* was measured once every day, and PDA without crude lipopeptides was the CK group. Three replicates were conducted in each group to determine the impact of crude lipopeptides on *A. alternata* growth. To evaluate the inhibitory effect, the fungal colony’s diameter was evaluated. The difference in the mass of the mycelial agar plugs before and after the incubation was used for the biomass change. The inhibition rate was calculated by measuring the growth diameter and expressed as the following formula:Inhibition percentage (%) = [(b − b_0_)/b] × 100%(1)
where the values of b (mm) and b_0_ (mm) represent the radial growth diameter of *A. alternata* in the experimental group and the CK group, respectively.

#### 4.4.3. Stability Experiments of Crude Lipopeptides

The method of Rong et al. was used with some modifications [43]. A total of 5 mL of crude lipopeptides was placed at various temperatures (25 °C, 60 °C, and 120 °C) for 1 h and then mixed with PDA in a ratio of 1:3. After solidification, *A. alternata* was put in the center and inoculated for 5 d at 28 °C. The effect of different temperature treatments of crude lipopeptides on *A. alternata* was detected by measuring the diameter of *A. alternata*. Three replicates were made in each group.

The pH of the crude lipopeptides was adjusted to 1.0, 3.0, 5.0, 9.0, 11.0, and 13.0 with hydrochloric acid or sodium hydroxide. After 4 h, the crude lipopeptides’ pH was readjusted to 7.0 and mixed with the PDA at a ratio of 1:3. After solidification, *A. alternata* was placed at the center of the medium and inoculated at 28 °C for 5 days. The impacts of distinct pH values on the *A. alternata* were detected by measuring the diameter of *A. alternata*. Three replicates of each pH treatment were used.

#### 4.4.4. Effect of Crude Lipopeptides on the Mycelium and Spores of *A. alternata*

A 7-day-old *A. alternata* with a diameter of 5 mm was placed at the center of the PDA, and the cell slide was then placed at a distance of 2 cm from the *A. alternata*. When the mycelium had grown halfway to the cell slide, it was taken out and placed in the liquid medium containing 1 mL of crude lipopeptides and 1 mL of PDB medium for 24 h, with sterile deionized water as the CK group. Each group was subjected to three replicates.

A 5 mL centrifuge tube containing 1 mL of *A. alternata* suspension of spores with a concentration of 1 × 10^5^ spore mL^−1^ was centrifuged for 6 min at 8000× *g* at 20 °C. The supernatant was discarded, and 1 mL of crude lipopeptides and 1 mL of PDB were added for 24 h, with sterile deionized water as the CK group. Three replicates were conducted for each group. After being rinsed three times with phosphate-buffered saline (PBS), the treated hyphae were placed in 2.5% glutaraldehyde fixed at 4 °C over 24 h. The hyphae were then immobilized for 1 h using 1% osmium tetroxide, followed by three PBS washes at 0.1 M concentration. Hyphae were dehydrated using 30%, 50%, 70%, 80%, 95%, and 100% ethyl alcohol in turn after being immobilized. Ultimately, upon gold coating and critical point drying, the hyphae adhered to the metal sheet and were used for scanning electron microscopy (SEM).

#### 4.4.5. Effects of Crude Lipopeptides on Cell Membrane Integrity of *A. alternata*

The cell membrane integrity experiment was carried out according to Hu et al. [47]. A total of 1 mL of *A. alternata* spore suspension with 1 × 10^5^ spore mL^−1^ in a 1.5 mL-centrifuge tube was centrifuged at 20 °C for 6 min at 8000× *g* and added to 1 mL of crude lipopeptides. After 3 h of incubating at 28 °C, the spore precipitate was washed with 1 mL of PBS and centrifuged for 6 min. Then, 1 mL of PBS and 20 μL of PI staining solution were immediately added. After 40 min at 28 °C in darkness, centrifugation was conducted, and the supernatant was discarded. Finally, 1 mL of PBS was added. Three runs of the experiment were carried out, with sterile water treatment acting as the control. *A. alternata* spores were observed and photographed with a fluorescence microscope, and cell mortality was calculated using a computer and expressed by the following equation:Cell mortality (%) = number of fluorescent spores/total amount of spores × 100%(2)

#### 4.4.6. Determination of Mycelial Conductivity

Sterile cellophane was carefully spread on the surface of the PDA medium, and then *A. alternata* was inoculated at 28 °C at the center of the medium above for 7 days. The cellophane was removed, and the mycelium on it was collected. Then, 0.25 g of mycelium and 5 mL of crude lipopeptides was added in a sterile centrifuge tube for incubation at 28 °C. The conductivity of the suspension was measured continuously with a conductivity meter at 0, 30, 60, 90, 120, 150 min, and 180 min, and the experiments were repeated three times.

#### 4.4.7. Effect of Crude Lipopeptides on the Content of *Alternaria* Toxins

A total of 6 mL of crude lipopeptides and 6 mL of *A. alternata* spore suspension (1 × 10^5^ spore mL^−1^) were combined with 6 mL of PDB medium. The control group consisted of PDB containing only 6 mL of *A. alternata* (1 × 10^5^ spore mL^−1^). After that, the culture was grown over 7 days at 28 °C in darkness using a rotary shaker incubator at 150 rpm/min.

The QuEChERS was used to extract *Alternaria* toxins [54]. The liquids in the media in the above treatment and CK group were vortexed for 15 s and mixed evenly. During extraction, 200 µL of the supernatant was mixed in 400 µL for acidic ethyl acetate after 1 mL of the above solution was centrifuged at 8000× *g* for 6 min. After centrifuging at 8000× *g* for 2 min, the supernatant was evaporated at 40 °C under a nitrogen stream until it was nearly dry. After 0.8 mL of the mixture (70:29:1, *v*/*v*/*v*) of acetonitrile, methanol, and formic acid was added, the residue was vortexed and filtered through a 0.22 µm filter to detect toxins using ultra-high-performance liquid chromatography–tandem mass spectrometry (UPLC-MS/MS). We calculated the percentage reduction in *Alternaria* toxins concentration using the following equation:The percentage reduction in *Alternaria* toxins concentration (%) = [(C_0_ − C)/C_0_] × 100%(3)
where C_0_ is the concentration of the *Alternaria* toxin in the CK group, and C is the concentration of the *Alternaria* toxin in the experimental group.

#### 4.4.8. Effect of Crude Lipopeptides on *A. alternata* and Toxins in Tomatoes

The postharvest biocontrol trials were conducted based on Estiarte et al.’s studies [55]. After sanitizing the processing tomato with 0.01% sodium hypochlorite for 2 min, it was rinsed three times with sterile water and allowed to air-dry at room temperature. Each tomato’s equator was punctured with a dissecting needle, measuring 1 mm in depth and 1.5 mm in width. Then, 10 µL of the crude lipopeptides was injected into the wounds. As a reference, 10 µL of distilled water was used. After half and hour, 10 µL of *A. alternata* (1 × 10^5^ spore mL^−1^) was inoculated into every wound. The tomatoes were then kept in plastic boxes at 28 °C for 7 days, during which time the diameters of the lesions were measured every day. Each replicate consisted of five tomatoes, and tests were conducted three times. By measuring the diameter of tomato spots, the suppression effect on fungal development was evaluated and represented by the following equation:Inhibition percentage (%) = [(d_0_ − d)/d_0_] × 100%(4)
where the values of d (mm) and d_0_ (mm) represent the radial growth diameter of *A. alternata* in the experimental group and the CK group, respectively.

On the 7th day, whole tomato fruits had been homogenized at room temperature. Then, 10 mL of acetonitrile containing 1% acetic acid and 5 mL of water were mixed with 5 g of finely homogenized tomato samples. To completely distribute the sample, the mixes were homogenized for 7 min at 2500 rpm using an automatic horizontal shaker. Then, 1 g of NaCl and 4 g of anhydrous MgSO_4_ were promptly added. After centrifuging at 8000× *g* for 5 min, the supernatant was evaporated at 40 °C under a nitrogen stream until nearly dry. After 0.8 mL of the mixture (70:29:1, *v*/*v*/*v*) of acetonitrile, methanol, and formic acid was added, the residue was vortexed and filtered through a 0.22 µm filter to detect toxins using UPLC-MS/MS [56].

#### 4.4.9. Detection of *Alternaria* Toxins Using UPLC-MS/MS

The Waters Acquity UPLC tandem quadrupole mass spectrometer, equipped with an Acquity UPLC HSS C18 column for separation, was used to detect *Alternaria* toxins. Eluents A and B of the mobile phase were methanol and 5 mM ammonium acetate, respectively. A gradient elution was used with an 8 min run duration as follows: 20% A for the first 0–1 min, increasing to 100% after 4 min and maintaining for 1.5 min; then, 20% A was used for the next 0.5 min and maintained for 2 min before column re-equilibration. Throughout the chromatographic separation, a flow rate of 0.3 mL/min was maintained, and an injection volume of 5 μL was used. At a capillary voltage of 2.5 kV, a desolvation temperature of 600 °C, a source block temperature of 125 °C, a desolvation gas of 1000 L h^−1^, and a cone nitrogen gas flow of 150 L h^−1^, the MS/MS analysis was run in the negative mode. Every *Alternaria* toxins’ ion chromatogram was acquired using the full-scan chromatogram’s MS^2^ mode [51]. The mass spectrometry parameters used by four *Alternaria* toxins are displayed in Table 1.

#### 4.4.10. Real-Time Quantitative Polymerase Chain Reaction Analysis (qRT-PCR)

The qRT-PCR experiment was carried out with samples from crude lipopeptides; treated and control group Hyphae samples were gathered after 7 days and triturated in liquid nitrogen to extract fungal RNA using an Omega fungal RNA Kit. The extracted RNA was measured for absorbance at 260 nm and 280 nm using a nanophotometer. RNA was reverse-transcribed to cDNA using the FastKing RT Kit, and a StepTwoPlus Real-Time PCR System was utilized for the quantitative real-time PCR using SuperReal PreMix Plus. The following were the settings for the qRT-PCR: 95 °C for 15 min, followed by 40 cycles of 95 °C for 15 s and 60 s. An internal control, the *AaActin* gene, was employed to standardize the expression data. Table 2 provides the gene-specific primers of the synthetases associated with *Alternaria* toxins. Each sample included three biological replicates, and the relative expression level was determined using the delta-delta Ct method (2^−∆∆Ct^ method).

### 4.5. Statistical Analysis

The experimental data were statistically calculated using Microsoft Office Excel 2017 and plotted using Origin 2021 (9.8.0.200) software. SPSS 18.0 was used to conduct the statistical analysis. The effects of the treatments were assessed using analysis of variance (ANOVA). The means were compared using Duncan’s multiple-range tests. *p* < 0.05 was used to indicate significant differences.

## Figures and Tables

**Figure 1 toxins-16-00065-f001:**
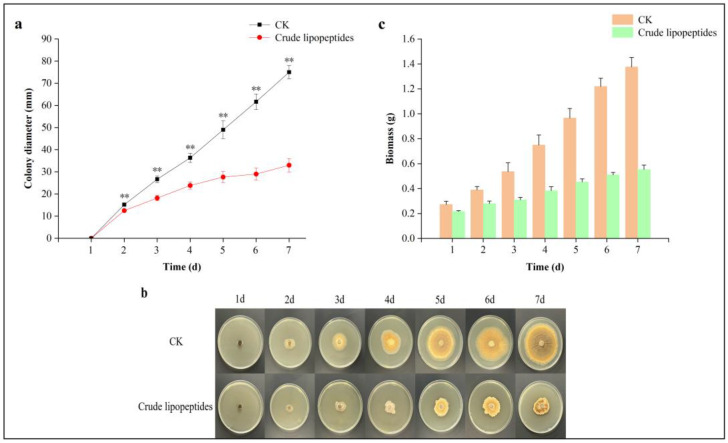
Inhibition of crude lipopeptides on the growth of *A. alternata* isolate H10 on PDA mediums after 7 days of incubation. (**a**) Diameter of mycelial growth within 7 days of incubation in the treatment and CK. (**b**) Mycelial growth in the treatment and CK group within 7 days of incubation. (**c**) Biomass of mycelial growth within 7 days of incubation in the treatment and CK. Asterisks indicate significant differences between treatments and CK (** *p* < 0.01). Mean ± SD, n ≥ 3.

**Figure 2 toxins-16-00065-f002:**
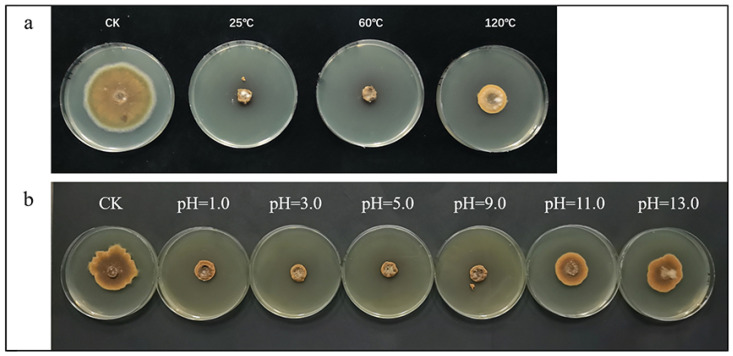
Effect of different temperature (**a**) and pH (**b**) treatments on the anti-fungal activity of crude lipopeptides.

**Figure 3 toxins-16-00065-f003:**
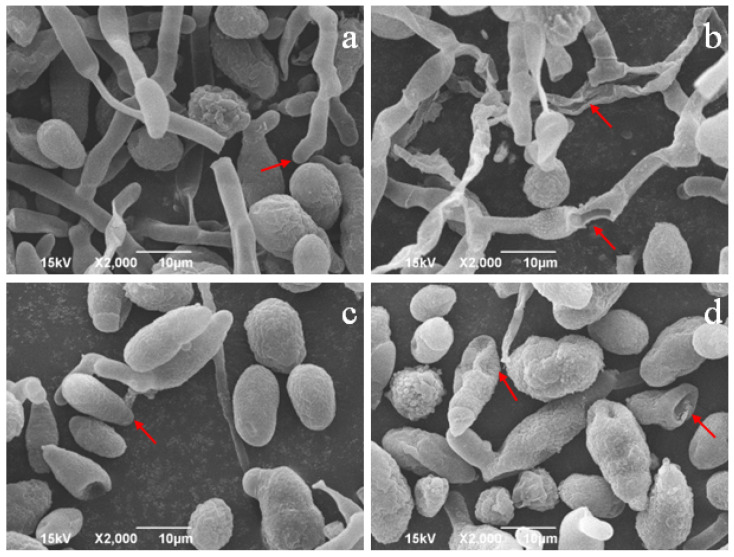
Scanning electron microscope of the mycelial in CK group (**a**) and treatment group (**b**). SEM micrograph for the spores in CK group (**c**) and treatment group (**d**). The red arrows refer to the changes in spores and mycelium.

**Figure 4 toxins-16-00065-f004:**
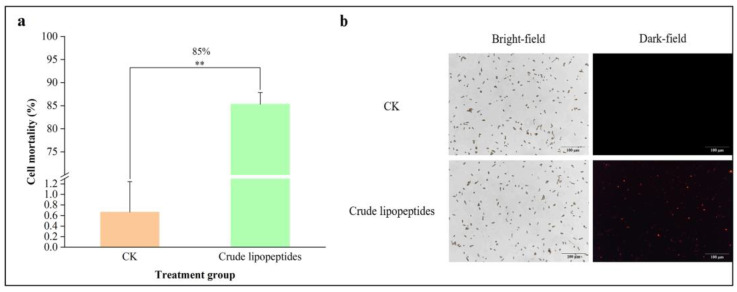
Effect of crude lipopeptides on the membrane integrity of *A. alternata* spore cells. (**a**) *A. alternata* spore cell membrane integrity. (**b**) PI staining of *A. alternata* spore cell membrane integrity. Asterisks indicate significant differences between treatments and CK (** *p* < 0.01). Mean ± SD, n ≥ 3.

**Figure 5 toxins-16-00065-f005:**
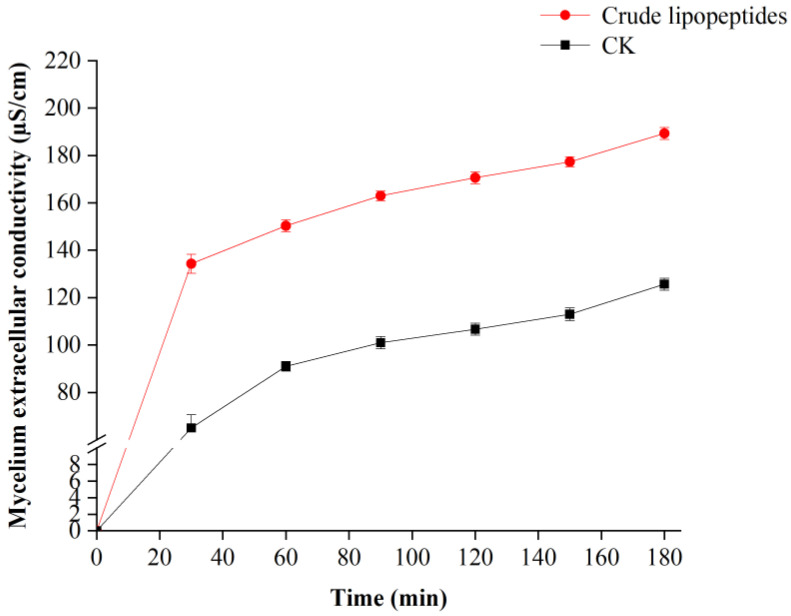
Effect of crude lipopeptides on the extracellular conductivity of *A. alternata* mycelium. Mean ± SD, n ≥ 3.

**Figure 6 toxins-16-00065-f006:**
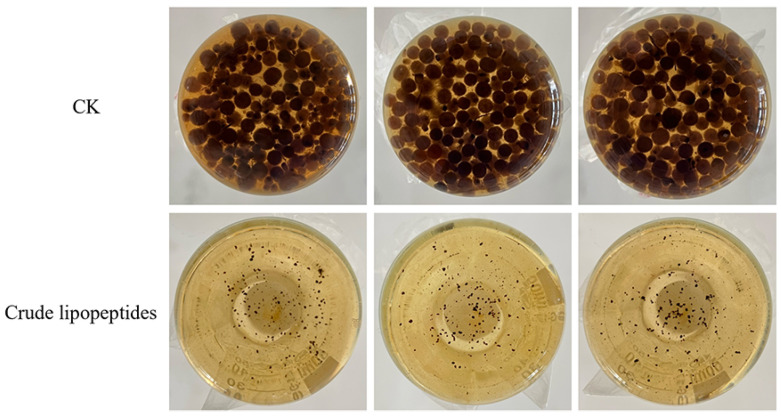
Anti-fungal activity of crude lipopeptides on PDB medium against *A. alternata*.

**Figure 7 toxins-16-00065-f007:**
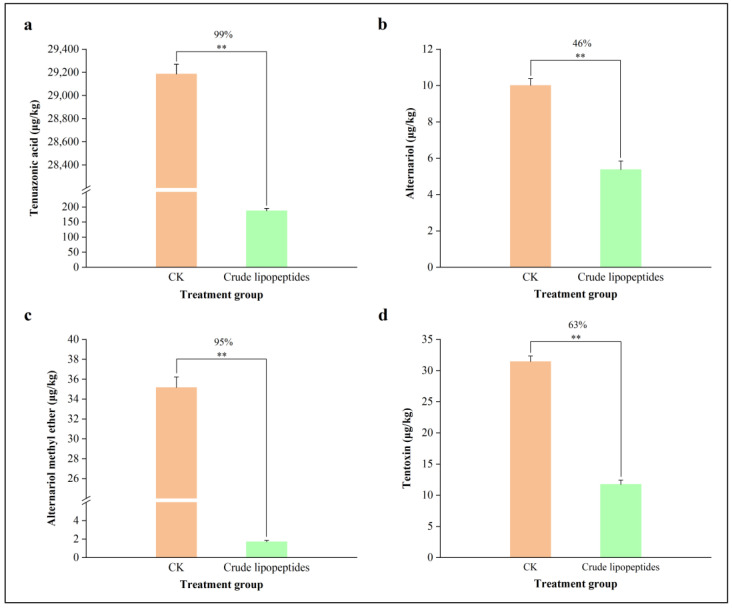
Inhibition of *Alternaria* toxins content by crude lipopeptides on PDB liquid medium. (**a**) TeA, (**b**) AOH, (**c**) AME, (**d**) TEN. Asterisks indicate significant differences between treatments and CK (** *p* < 0.01). Mean ± SD, n ≥ 3.

**Figure 8 toxins-16-00065-f008:**
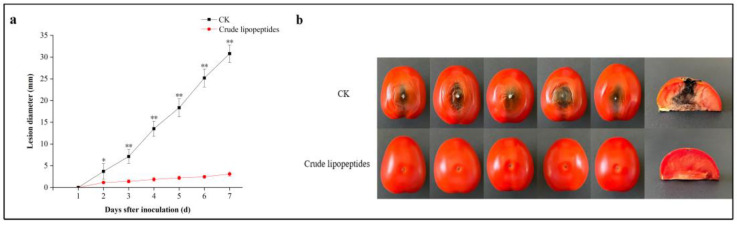
Anti-fungal potential of crude lipopeptides on tomato. (**a**) The lesion diameter in the treatment and CK during 7 days of storage. (**b**) The black spot disease symptoms in tomatoes in the treatment and CK at 7 days. Asterisks indicate significant differences between treatments and CK (* *p* < 0.05, ** *p* < 0.01). Mean ± SD, n ≥ 3.

**Figure 9 toxins-16-00065-f009:**
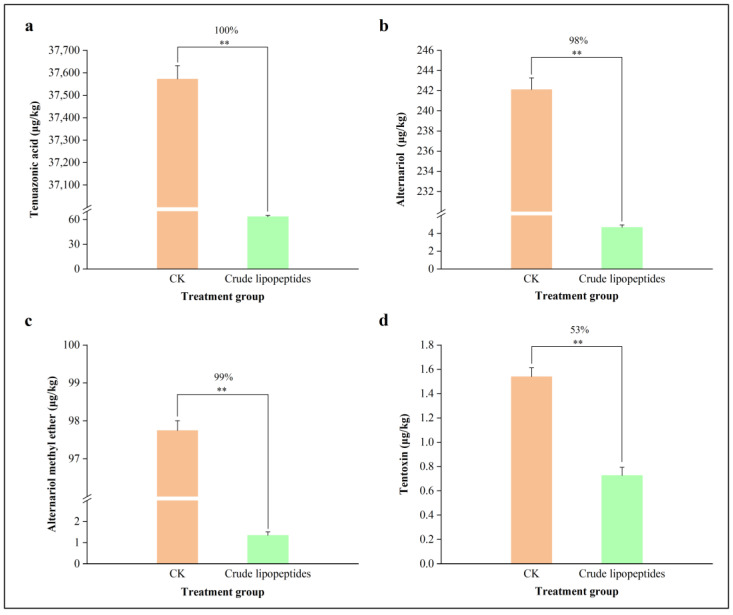
Inhibition of *Alternaria* toxins content by crude lipopeptides in tomato. (**a**) TeA, (**b**) AOH, (**c**) AME, (**d**) TEN. Asterisks indicate significant differences between treatments and CK (** *p* < 0.01). Mean ± SD, n ≥ 3.

**Figure 10 toxins-16-00065-f010:**
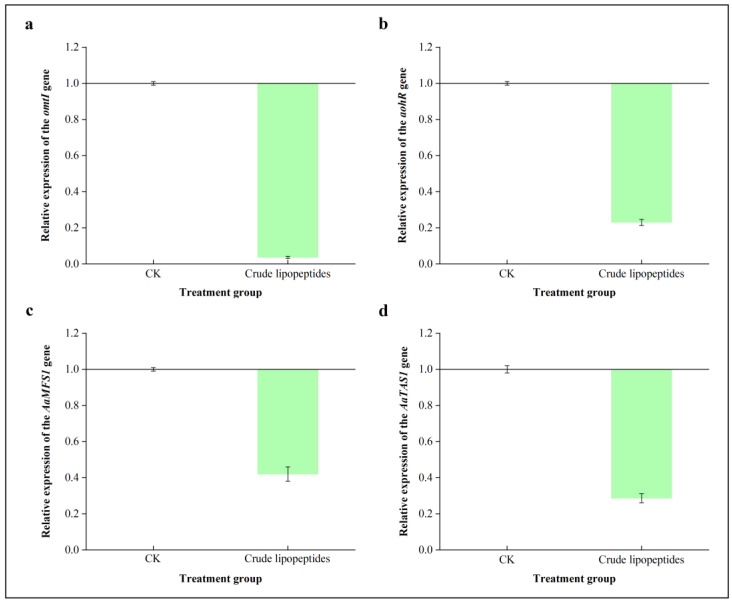
Effects of crude lipopeptides on the expressions of the critical genes involved in *Alternaria* toxins biosynthesis. (**a**) *omtI*, (**b**) *aohR*, (**c**) *AaMFS1*, (**d**) *AaTAS1.* Mean ± SD, n ≥ 3.

**Figure 11 toxins-16-00065-f011:**
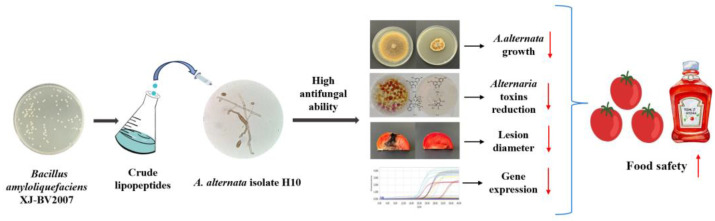
Possible inhibitory pathways of *A. alternata* isolate H10 by crude lipopeptides. The red arrows indicate rising and falling respectively.

**Table 1 toxins-16-00065-t001:** The LC-MS/MS parameters for *Alternaria* toxins.

Analyte	Precursor Ion (m/z)	Product Ions (m/z)	Retention Time (min)	Collision Energy (eV)	Cone Voltage (V)
TeA	196.1	138.9 *	1.80	18	30
		196.0		15	
AOH	257.0	213.2 *	3.98	16	43
		147.1		14	
AME	271.1	256.1 *	5.42	17	46
		228.2		15	
TEN	413.1	141.2 *	5.00	19	30
		214.2		13	

* The product ion content is the highest and is, therefore, used to quantify.

**Table 2 toxins-16-00065-t002:** Sequence information of key genes related to production of *Alternaria* toxin.

No.	Gene Name		Sequences (5′→3′)	References
1	*AaActin*	F′	GCAGCATGTACCCAGGTCTT	[9]
		R′	GGATCTTCGATGCGGACCTT	
2	*omtI*	F′	CAAGATCCCAAAGTCAAAGGA TGGTGGCCTACACTCTAATGG	[10,37]
		R′	GACGTCATATAATCATACGGC TAAGCCAGTGTTGCACCAATG	
3	*aohR*	F′	TCCTTATCCTGGACGACAT	[37]
		R′	GAGGTTGATGACGGCTTC	
4	*AaMFS1*	F′	CCCCGGTGCTGTTGATGTGGATAG	[9]
		R′	TCGGACGCAATGGAGAGGAAGAGC	
5	*AaTAS1*	F′	GGTCGAGCAGTCAAACCTGA	[9]
		R′	GTCGATGACAGTCGCGAGAT	

## Data Availability

Data are contained within the article.

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
