# Peer review of "Crude Lipopeptides Produced by Bacillus amyloliquefaciens Could Control the Growth of Alternaria alternata and Production of Alternaria Toxins in Processing Tomato"

_toxins, 2024, doi:10.3390/toxins16020065_

Round 1

Reviewer 1 Report

Comments and Suggestions for Authors

The goals of the present investigation were to assess the impact of crude lipopeptides from B. amyloliq- uefaciens XJ-BV2007 upon the control for A. alternata isolate H10 and Alternaria toxins with in vivo and in vitro, and to explore the possible mechanisms of crude lipopeptides in con- trolling the postharvest black spot diseases. Below are a few questions and comments that may help improve the quality of the content presented:

1. According to the method in 4.4.7, the results obtained in 2.3 mention that the accumulation of TeA, AOH, AME, and TEN decreased by 99%, 46%, 95%, and 63%, respectively. How is the formula for the source of this numerical result derived?

3. It is suggested to replace the section "2.7 Possible inventory path to Alternative toxins" with a discussion section, which provides a detailed analysis of inventory path to Alternative toxins.

3. In section 4.4.1, a series of operations such as diluting XJ-BV2007 fermentation broth were carried out to obtain a final yield of 2ml. What is the amount of sample that can yield 2ml?

4. It is recommended to use a unified Roman numeral representation for the numbers appearing in the article, such as "five ML" in 4.4.2.

 5.The effect of crude lipopeptide on the integrity of the cell wall of streptospora should be included in chapter 1.

6. The accumulation calculation method in 4.4.7 does not insert formulas or reference literature.

7. There is no uniform use of numbers in the article, not all use Roman numerals.

Author Response

Response to Reviewer Comments

Dear Editors and Reviewers:

Thank you for your letter. We thank the reviewers again for the time and effort that they have put into reviewing the previous version of the manuscript. Those comments are all valuable and very helpful for revising and improving our paper, as well as the important guiding significance to our researches.

We are very grateful to your comments for the manuscript. According with your advice, we tried our best to amend the relevant part and made some changes in the manuscript. These changes will not influence the content and framework of the paper. All of your questions were answered below. And here we list the changes and marked in red in revised paper. Please see the attachment for point to point response.

We appreciate for Reviewers’ warm work earnestly, and hope that the correction will meet with approval. Should you have any questions, please contact us without hesitate. Please see the attachment.

Once again, thank you very much for your comments and suggestions.

Reviewer 2 Report

Comments and Suggestions for Authors

The manuscript details the research and the results obtained for using crude lipopeptides produced by Bacillus amyloliquefaciens to control the growth of Alternaria alternata and the production of mycotoxins in processing tomatoes.

Research has addressed:

-     Effect of crude lipopeptides treatment on mycelial growth of Alternaria alternata;

-     Stability test of antifungal activity of crude lipopeptides;

-    Inhibition mechanism of crude lipopeptides on Alternaria alternata (mycelium and spores microstructure; membrane integrity of spore cells; conductivity of mycelium; content of Alternaria toxins; A.  alternata and its toxins on tomato; expression of key genes related to Alternaria Toxins).

Materials and Methods are presented in detail. The Results, Disscutions and Conclusions showed that lipopeptides produced by Bacillus amyloliquefaciens have the following effects on Alternaria alternata in tomatoes: -          Inhibitory effects on the growth of Alternaria alternata mycelia and spores; -          Inhibitory effects on mycotoxin synthesis: Tenuazonic acid (TeA), Alternariol (AOH), Alternariol methyl ether (AME) and Tentoxin (TEN); -          Lipopeptides damaged the integrity and permeability of spore cell membranes; -          Inhibitory effects on the growth of Alternaria alternata spore and the diameter of tomato spots and the accumulation of Alternaria toxins was reduced in vivo and in vitro.

I consider that the manuscript requires polishing the text and addressing the following aspects before being published.

1. Title: ’’Crude lipopeptides produced by Bacillus amyloliquefaciens could control the growth of Alternaria alternata and production of Alternaria toxins in tomato processing’’

2. Abstract. Alternaria spp. and its toxins, are is the main pollutant contaminants of processing tomatoes in Xinjiang,

- Use ‘’contaminant’’ instead of ‘’pollutant’’ (L6) and ‘’contaminated’’ instead of ‘’tainted’’ (L49).

3. Tomato  (Solanum lycopersicum) Cultivation areas and environmental conditions. Top 10 tomato producers: China, India, USA, Turkey, Egypt, Iran, Italy, Spain, Mexico and Brazil.

https://www.atlasbig.com/en-gb/countries-by-tomato-production

4. Tomato plants will grow well in well-drained sites that receive full sun for most of the day. The soil pH should be slightly acidic (6.2 to 6.8).

https://extension.unh.edu/resource/growing-vegetables-tomatoes-fact-sheet-1#:~:text=Tomato%20plants%20will%20grow%20well,foliage%20but%20little%20fruit%20production.

5. Use soil pH (6.2 to 6.8) in Discussions (L141)At pH 1.0 to 9.0, the activity of crude lipopeptides 142 was unaffected, while at pH 11.0 to 13.0, the activity of crude lipopeptides decreased and 143 the A. alternata radial diameter increased (Figure 2b).

6. Xinjiang Province is located at 41°N 85°E in northwest China. It has grey desert soil and aeolian sandy soil as the primary agricultural soil types, and semiarid or desert climate with great seasonal differences in temperature.

- Dong X, Zhang Z, Wang S, Shen Z, Cheng X, Lv X, Pu X. 2022Soil properties, root morphology and physiological responses to cotton stalk biochar addition in two continuous cropping cotton field soils from Xinjiang, ChinaPeerJ 10:e12928 https://doi.org/10.7717/peerj.12928

- VeroustraeteF.; LiQ.; VerstraetenW.W.; ChenX.; BaoA.; DongQ.; Liu T.; Willems, P. (2012)Soil moisture content retrieval based on apparent thermal inertia for Xinjiang province in China, International Journal of Remote Sensing, 33:12, 3870-3885, DOI: 10.1080/01431161.2011.636080

https://weatherandclimate.com/china/xinjiang

https://web.archive.org/web/20100906034159/http://koeppen-geiger.vu-wien.ac.at/survival

7. How the contamination of tomatoes with Alternaria alternata and mycotoxins can be influenced by Climate Changes (CC) in Xinjiang Province.

- Zhou G, Chen Y and Yao J (2023) Variations in precipitation and temperature in Xinjiang (Northwest China) and their connection to atmospheric circulation. Front. Environ. Sci. 10:1082713. doi: 10.3389/fenvs.2022.1082713

- Use this information in 2.2. Stability Test of Antifungal Activity of Crude Lipopeptides.

8. If possibleMultivariate Tests of BetweenSubjects Effects: Bacillus lipopeptides, Altenaria growth, mycotoxin production, temperature, pH etc.

9.  Environmental condition for Alternaria alternata growth and toxins contamination in tomatoes. Spores survive in soil and tomato contamination.

10.. (L74) ….mycotoxin synthesis  manufacturing.

11. Figures 1, 4, 7 …. Significant differences substantial variations between treatments and CK (* < 0.05, ** < 0.01).

12. (L171). Propidium iodide (PI) is a nuclear staining reagent ….

13. The delta-delta Ct method (2–∆∆Ct method).

14. (L319). Nutrient agar (NA) and nutrient broth agar (NB).

15. Cite the publicationQin Q, Fan Y, Jia Q, Duan S, Liu F, Jia B, Wang G, Guo W, Wang C. The Potential of Alternaria Toxins Production by Aalternata in Processing Tomatoes. Toxins (Basel). 2022 Nov 24;14(12):827. doi: 10.3390/toxins14120827. (Is it your publication?)

Comments on the Quality of English Language

I consider that the manuscript requires polishing the text.  

Author Response

Dear Editors and Reviewers:

Thank you for your letter. We thank the reviewers again for the time and effort that they have put into reviewing the previous version of the manuscript. Those comments are all valuable and very helpful for revising and improving our paper, as well as the important guiding significance to our researches.

We are very grateful to your comments for the manuscript. According with your advice, we tried our best to amend the relevant part and made some changes in the manuscript. These changes will not influence the content and framework of the paper. All of your questions were answered below. And here we list the changes and marked in red in revised paper. Please see the attachment for point-to-point responses.

We appreciate for Reviewers’ warm work earnestly, and hope that the correction will meet with approval. Should you have any questions, please contact us without hesitate. Please see the attachment.

Once again, thank you very much for your comments and suggestions.

Reviewer 3 Report

Comments and Suggestions for Authors

The manuscript present data on the efficiency of a crude lipopeptides extract from a culture broth of a selected bacterium for controlling Alternaria alternate and its mycotoxins contamination, both in vitro and on tomatoes. The strengths are the various methods used to identify how the product could act on the fungus. The major weakness is the fact that only one concentration was tested and it is difficult from the Materials and Methods to know exactly what is was. A discussion on the feasibility of using such extracts under agronomic conditions would be appreciate, since in the manuscript the crude lipopeptides and the fungus were injected into created wounds in tomato fruits. A second weakness is the lack of chemical characterization of the crude lipopeptides extracts.

Specific comments:

 Line 79: I do not understand “Based on the variability of genetic manipulators….” May be manipulator is not the right name.

Lines 100-101: “…with in vivo and in vitro, and…” May be a word is missing. I suggest ‘… with in vitro and in vivo assays, and…’

Line 131: Asterix are only for fig 1a, in Fig1b, different letters. You should add information or change the graphs. For all the figures, add information on what are the error bars (SD of x replicates?)

Line 160: alternata

Lines 195-196: Actually, the difference is the same at 0 min, showing an instant effect of crude lipopeptides.

Line 200: You should add ‘immediately’.

Line 244: ‘tomato toxins’ means that toxins are produced by tomatoes. You should change.

Lines 269-270: Why this reference here, it is not Alternaria toxins. It could be deleted.

Line 292 – Figure 11 doesn’t show “Possible inhibitory pathways of A. alternata isolate H10 by crude lipopeptides”. Actually, it in just a graphical abstract of the manuscript and is not informative here.

Line 352: information missing to allow to know what are the concentrations used. Were there any dilutions. If not, you should have obtained large quantities of extracts. As in line 346  you wrote ‘Its weight was then determined…’ you should be able to give information on concentrations.

Author Response

Dear Editors and Reviewers:

Thank you for your letter. We thank the reviewers again for the time and effort that they have put into reviewing the previous version of the manuscript. Those comments are all valuable and very helpful for revising and improving our paper, as well as the important guiding significance to our researches.

We are very grateful to your comments for the manuscript. According with your advice, we tried our best to amend the relevant part and made some changes in the manuscript. These changes will not influence the content and framework of the paper. All of your questions were answered below. And here we list the changes and marked in red in revised paper.

We appreciate for Reviewers’ warm work earnestly, and hope that the correction will meet with approval. Should you have any questions, please contact us without hesitate. Please see the attachment.

Once again, thank you very much for your comments and suggestions.
